neuroscience/genetics/physiology

*bdnf*, *pcna*, neurogenesis, Atlantic salmon, hippocampus, dorsolateral pallium

**Author for correspondence:**
Marco A. Vindas
e-mail: marco.vindas@nmbu.no

# Swimming exercise enhances brain plasticity in fish

Daan Mes[1], Arjan P. Palstra[4], Christiaan V. Henkel[2], Ian Mayer[1] and Marco A. Vindas[3,5]

[1]Department of Production Animal Clinical Sciences, [2]Department of Basic Sciences and Aquatic Medicine, and [3]Department of Food Safety and Infection Biology, Norwegian University of Life Sciences, Oslo, Norway
[4]Wageningen University and Research Animal Breeding and Genomics, Wageningen Livestock Research, Wageningen, The Netherlands
[5]Uni Environment, Uni Research AS, Bergen, Norway

DM, 0000-0001-9704-6897; MAV, 0000-0002-3996-0952

It is well-established that sustained exercise training can enhance brain plasticity and boost cognitive performance in mammals, but this phenomenon has not received much attention in fish. The aim of this study was to determine whether sustained swimming exercise can enhance brain plasticity in juvenile Atlantic salmon. Brain plasticity was assessed by both mapping the whole telencephalon transcriptome and conducting telencephalic region-specific microdissections on target genes. We found that 1772 transcripts were differentially expressed between the exercise and control groups. Gene ontology (GO) analysis identified 195 and 272 GO categories with a significant overrepresentation of up- or downregulated transcripts, respectively. A multitude of these GO categories was associated with neuronal excitability, neuronal signalling, cell proliferation and neurite outgrowth (i.e. cognition-related neuronal markers). Additionally, we found an increase in *proliferating cell nuclear antigen* (*pcna*) after both three and eight weeks of exercise in the equivalent to the hippocampus in fish. Furthermore, the expression of the neural plasticity markers *synaptotagmin* (*syt*) and *brain-derived neurotrophic factor* (*bdnf*) were also increased due to exercise in the equivalent to the lateral septum in fish. In conclusion, this is the first time that swimming exercise has been directly linked to increased telencephalic neurogenesis and neural plasticity in a teleost, and our results pave the way for future studies on exercise-induced neuroplasticity in fish.

## 1. Introduction

There is ample evidence in the mammalian literature that exercise training leads to increased neurogenesis and synaptic plasticity and that this is associated with increased cognition. Specifically, running exercise has been shown to improve the cognitive

performance of rodents in spatial tasks [1,2]. This effect is strongly associated with increased neurogenesis and synaptic plasticity in the hippocampus, particularly in the dentate gyrus, mediated by an increased abundance of growth factors, neurotransmitters and neurotrophic factors, although the exact mechanisms are yet to be elucidated [2–4].

Even though the link between exercise, neural plasticity and cognition is now well-described in mammals, this phenomenon has not received much attention in other vertebrates, such as fish. In this context, fish species are a promising model to study neurobiological mechanisms, particularly those associated with neurogenesis, since fish display neurogenesis in a multitude of proliferation zones throughout their entire lives and rates of cell proliferation in the teleost brain are one to two orders of magnitude higher than in the mammalian brain [5]. These higher cell proliferation rates, besides imparting remarkable neural plasticity, also contribute to the fact that upon neural damage, teleost fish species have an incredible capacity for regeneration of the central nervous system [6]. Notably, a pilot study conducted by Luchiari & Chacon [7] demonstrated that exhaustive swimming exercise in zebrafish (Danio rerio) improved their learning performance in a conditioning test within several days. Furthermore, a 10-day swim training regime promoted the expression of neurogenesis markers in the brain of zebrafish larvae [8].

In contrast to mammals, fish lack a six-layered pallium, but nonetheless, they are able to display a number of higher cognitive functions, which are mainly under forebrain telencephalic control [9–11]. Importantly, the fish telencephalon contains neural populations and networks associated with emotional and relational memory, learning and stress-reactivity, thus driving processes which show functional resemblances to processes which are under limbic system control in mammals [12–15]. Therefore, we hypothesize that increased exercise training will enhance telencephalic neurogenesis and neural plasticity in fish.

The aim of this study was to assess whether sustained swimming exercise can promote forebrain neurogenesis and neuroplasticity in Atlantic salmon (Salmo salar, L.). We chose salmon since this species is characterized by sustained periods of increased swimming exercise (i.e. migrations to and from seawater to freshwater spawning grounds [16]), and swimming exercise has already been shown to have beneficial effects in salmon, including increased growth rates [17] and stress alleviation [18]. We subjected fish to an eight-week sustained swimming exercise regime and assessed their expression of neural plasticity and neurogenesis markers at three and eight weeks. We here report that swimming exercise leads to the upregulation of key neuroplasticity- and neurogenesis-related genes in telencephalic areas of the salmon brain, similarly to effects reported in mammalian studies. These results highlight a promising new model for the study of swimming enhanced neurogenesis/neuroplasticity mechanisms and their link to enhanced cognition.

# 2. Material and methods

## 2.1. Ethics statement

This experiment was performed in accordance with Dutch law for experimentation and procedures on live animals. The experimental protocol was approved by the Animal Experimental Committee (DEC) of Wageningen University and Research (case no. 2016.D-0039).

## 2.2. Experimental fish

Experimental fish were hatchery-reared Atlantic salmon juveniles, which were first-generation offspring from wild-caught parents from the river Imsa, in southwestern Norway. Eggs hatched in late January 2017 and fish were reared under standard hatchery conditions at the Norwegian Institute for Nature Research (NINA) Research Station at Ims, Norway, in water from the adjacent river Imsa. At eight months old, fish were transported from the Ims hatchery to the aquaculture research facilities at Wageningen University and Research (WUR), the Netherlands. After 18 days of acclimatization at the WUR experimental facilities, fish were tagged intraperitoneally with passive integrated transponders tags (Trovan ID100A/1.4 mini transponders) and the fish were then left to recover for an additional week before the experiment started. Fish were 9-months-old at the start of the experiment and 11-months-old at final sampling.

## 2.3. Swimming exercise regime

Fish were exercised for a total of eight weeks on a volitional training regime (i.e. fish could, to a certain degree, choose their preferred swimming speed). At the start of the exercise regime, fish measured $123 \pm 5$ mm (fork length; FL) and weighed $20.8 \pm 3.6$ g (mean ± s.d.).

The experimental set-up consisted of two standard cylindrical 800-l holding tanks holding 110 fish each (density = 2.6 kg m$^{-3}$), of which the exercise treatment tank received a high-water flow adjacent to the tank wall. At the bottom of the exercise tank, the flow rate varied from 5 cm s$^{-1}$ or 0.4 body lengths (BL) s$^{-1}$ in the centre, to 27 cm s$^{-1}$ or 2.2 BL s$^{-1}$ at the outer wall. The flow rates at the water surface were 10 cm s$^{-1}$ or 0.8 BL s$^{-1}$ in the centre and 36 cm s$^{-1}$ or 2.9 BL s$^{-1}$ at the outer wall. Thus, by positioning themselves throughout the tank, fish could 'choose' their preferred swimming speed. Water flow in the control tank was less than 5 cm s$^{-1}$ or 0.4 BL s$^{-1}$ throughout the tank. The selected flow rates were the maximum speeds that could be achieved in the standard hatchery tanks and were well within the aerobic scope of salmon [19], as well as within the preferred range of flow rates of Atlantic salmon in natural habitats [20]. Both tanks were covered with mesh and half of the tank was covered with black foil to provide shelter. Light intensity at the water surface was approximately 45 lux. Exercised fish showed no sign of fatigue and generally positioned themselves facing the current, while occasionally drifting down with the current.

The light cycle was maintained at 12 L : 12 D throughout the experiment. Water temperature was maintained at 14.9 ± 0.45°C and nutrient levels were 0.06 ± 0.05 mg NH$_4$ l$^{-1}$, 0.08 ± 0.04 mg NO$_2$ l$^{-1}$ and 67.6 ± 24 mg NO$_3$ l$^{-1}$ (mean ± s.d.). Fish were fed commercial pellets (Nutra Parr, Skretting, Stavanger, Norway) by hand, twice per day until satiation and water flow was stopped during feeding to provide equal feeding opportunities for both exercised and sedentary fish. All fish were measured and weighed after the swimming treatment and specific growth rates were calculated as follows:

$$SGR = \ (\ln{(BM_f)} - \ln{(BM_i)}) \times \frac{100}{t},$$

where BM$_f$ is the final body mass in g at the end of the exercise period, BM$_i$ is the initial body mass in g at the start of the experiment and $t$ is the experimental time in days.

## 2.4. Sampling

Fish were sampled at two time points: after three and eight weeks for microdissection and at eight weeks only for RNAseq analysis. Fish were randomly collected from their holding tanks and quickly anaesthetized in 2-phenoxyethanol (VWR #26244.290, 1.3 ml l$^{-1}$). Opercular movement ceased completely within 30 s, after which weight and length were recorded. Immediately after, fish were decapitated and brains were sampled in two ways: (i) the jaw and gills were trimmed before fish heads were sealed in a plastic bag, snap-frozen on dry ice and stored at −80°C until processing or (ii) brains were dissected out and the telencephalon was extracted out and placed overnight in RNAlater (Invitrogen AM7024) at 4°C. The following day, surplus RNAlater was removed and samples were stored at −80°C.

## 2.5. Microdissections

Frozen trimmed skulls of 10 fish per treatment and per time period were sectioned (100 µm thick) transversely in a cryostat (Leica CM 3050) at −22°C. Sections were thaw-mounted onto glass slides (VWR 631-151) and subsequently stored at −80°C. Microdissections were performed on frozen sections kept on a cooling plate (−14°C) as described by Vindas *et al*. [21]. Three subregions of the telencephalon were microdissected: the dorsolateral pallium (Dl), the dorsomedial pallium (Dm) and the ventral part of the ventral telencephalon (Vv). On average, per individual, a total of 21 punches were taken for the Dl and the Dm and 11 for the Vv. Microdissected tissue was injected into RLT buffer (RNeasy Plus Micro Kit, Qiagen 74034) and immediately frozen at −80°C until RNA extraction, which was conducted within 3 days after microdissection.

## 2.6. Relative transcript abundance

Relative transcript abundance of *brain-derived neurotrophic factor (bdnf)*, *neural differentiating factor (neurod)*, *synaptotagmin (syt)* and *proliferating cellular nuclear antigen (pcna)* in microdissected areas was measured using real-time PCR (qPCR). Microdissected tissue was thawed, vortexed for 30 s, centrifuged at 13 400×g for 5 min and total RNA was subsequently extracted using the RNeasy Micro Plus Kit (Qiagen, ID 74034), according to the manufacturer's instructions. RNA concentrations were measured using nanodrop and the quality of extracted RNA was checked on a subset of samples using a Bioanalyzer RNA 6000 Pico Kit (Agilent 2100): RNA integrity numbers (RIN) were 8.5 ± 0.8 (mean ± s.d.). Reverse transcription was performed using an iScript cDNA Synthesis Kit (Bio-Rad 1708891) according to the manufacturer's instructions, using 90 ng of total RNA as template in a total reaction volume of 20 µl. Subsequently, cDNA was stored at −20°C.

The four target genes, as well as three reference genes (*elongation factor 1αa* (*ef1αa*), *ribosomal protein s20* (*s20*) and *hypoxanthine phosphoribosyltransferase 1* (*hprt1*)) were selected for qPCR. Previously published primer sequences (electronic supplementary material, table S1) were retrieved from the National Center for Biotechnology Information (NCBI: http://ncbi.nlm.nih.gov/nuccore). Calibration curves were run for all primer pairs (electronic supplementary material, table S2) and qPCR products were sequenced to confirm the specificity of the primers. The stability of the three reference genes *ef1αa*, *s20* and *hprt1* was evaluated (following protocol by Silver *et al*. [22]), after which *s20* was selected as the most stable reference gene.

Real-time PCR was carried out in duplicate using a Roche Light Cycler 96 (Roche Diagnostics, Penzberg, Germany) and accompanying software (version 1.1.0.1320). The reaction volume was 10 µl including 5 µl LightCycler® 480 SYBR® Green I Master (04887352001, Roche Diagnostics GmbH, Mannheim, Germany), 1 µl of each forward and reverse primer (1 nM final concentration for each primer) and 3 µl of cDNA (diluted 1 : 5). Cycling conditions were 10 min at 95°C, followed by 40 cycles of 10 s at 95°C, 10 s at 60°C and 8 s at 72°C, followed by a melting curve analysis. Samples which had Cq values greater than 35 and/or technical replicates with an s.d. difference greater than 0.1 were eliminated following the methodology of qPCR analysis by Bustin *et al*. [23]). A calibrator, made by pooling aliquots of cDNA of all samples, was included in triplicate in all plates to allow for comparison of Cq values between plates. Expression values were calculated according to Vandesompele *et al*. [24], and are expressed as relative to the expression of the reference gene *s20*. Gene expression levels of the three-week control fish were normalized to 1, and data are presented as normalized values to this treatment control average (fold-change).

## 2.7. RNA sequencing tissue processing and analysis

The telencephalon samples ($n = 4$ per treatment) were homogenized using a TissueRuptor (Qiagen, Venlo, The Netherlands) and total RNA was extracted using the miRNeasy mini kit (Qiagen, Venlo, The Netherlands) according to the manufacturer's instructions. Integrity and concentration of the RNA were checked on a Bioanalyzer 2100 total RNA Nano series II chip (Agilent, Amstelveen, The Netherlands) and the median RIN value was 9.0. Illumina RNA-seq libraries were prepared from 0.5 µg total RNA using the Illumina TruSeq® Stranded mRNA Library Prep kit according to the manufacturer's instructions (Illumina, San Diego, USA). All RNA-seq libraries (150–750 bp inserts) were sequenced on an Illumina HiSeq2500 sequencer as $1 \times 50$ nucleotides single-end reads according to the manufacturer's protocol. Image analysis and base calling were done using the Illumina pipeline.

The reads were aligned to the latest version of the Atlantic salmon genome reference (ICSASG v. 2, NCBI RefSeq GCF_000233375.1; [25]) using TopHat v. 2.0.13 [26] at 'very-sensitive' default settings. Samtools v. 1.2 [27] was used to remove secondary alignments, i.e. alignments that meet TopHat's reporting criteria but are less likely to be correct than simultaneously reported primary alignments. Alignments to annotated exons were counted and summarized at the gene level using HTSeq-count v. 0.10.0 using the 'intersection-nonempty' setting [28].

## 2.8. Differential expression analysis

Raw read counts for 48 436 protein-coding genes were analysed in R v. 3.4.4 (R Development Core Team 2016) using the edgeR package v. 3.20.9 [29]. Initially, read counts were normalized using the TMM method and a multi-dimensional scaling (MDS) plot was generated to identify outliers. After outlier removal, the read counts were normalized again, and differential expression between the four treatment groups was calculated using edgeR's recommended quasi-likelihood *F*-test for generalized linear models. MDS plots and differential expression were only calculated for genes with at least 10 aligning reads in each sample. For downstream analyses, reads per kilobase million (RPKM) expression values (normalized between samples and corrected for transcript length) were exported from edgeR. Transcripts with a false discovery rate (FDR) less than 0.01 were considered to be significantly differentially expressed between treatments. The TM4 MultiExperiment Viewer v. 4.9.0 (www.tm4.org) was used to visualize expression profiles in heat maps.

## 2.9. Gene ontology analysis

Gene ontology (GO) annotations for the ICSASG_v2 assembly were retrieved using the Ssa.RefSeq.db R package v. 1.2 (https://rdrr.io/github/FabianGrammes/Ssa.RefSeq.db/), and overrepresentation of

'Biological Process' categories was assessed using the R package GOseq [30], using the 'Wallenius' method and including correction for transcript length.

## 2.10. Statistical analyses

Body mass at the start of the experiment was compared using a Student's *t*-test. Two-way analysis of variance (ANOVA) was used to compare body mass and SGR, with treatment (training versus control) and time (three and eight weeks) as independent variables. The gene expression data included treatment, time and sex as explanatory variables. The most parsimonious model (with the lowest corrected Akaike Information Criterion score; AICc) was a model with only treatment and time as explanatory variable, and this model was subsequently used. Models were assessed by their capacity to explain the variability and interaction effects between treatment and conditions were accepted or rejected according to total model 'lack of fit' probabilities (provided by the ANOVA model). Differences between all groups were assessed by Tukey–Kramer honestly significant difference post hoc test. Before final acceptance of the model, diagnostic residual plots were examined to ensure that no systematic patterns occurred in the errors (e.g. fitted values versus observed values and *q*–*q* plots). Data outliers were determined by the $\chi^2$ test and eliminated when *r* values were greater than *r* tables (with 95% confidence). All data reached normality and are presented as mean ± s.e.m.

# 3. Results

## 3.1. Growth

There were no significant morphometric differences between experimental groups at the start of the experiment [$t_{(221)} = 0.97$, $p = 0.33$]. There was a significant effect of time [$F_{(3,203)} = 68.6$, $p < 0.001$] and a significant interaction between time and treatment [$F_{(3,203)} = 4.7$, $p = 0.03$], but no treatment effect [$F_{(3,203)} = 2.54$, $p = 0.11$] on body mass, with exercised fish having a higher body mass at eight weeks ($p = 0.002$), but not at three weeks ($p = 0.99$), compared to controls. In addition, there was a significant treatment [$F_{(3,203)} = 8.16$, $p = 0.005$] and time [$F_{(3,203)} = 334$, $p < 0.001$] but no interaction [$F_{(3,203)} = 0.29$, $p = 0.59$] effect on the SGR, with exercised fish having a higher SGR at eight weeks ($p = 0.007$), but not at three weeks ($p = 0.53$), compared to controls (figure 1).

## 3.2. Gene expression

### 3.2.1. Synaptotagmin

There was a significant effect of treatment [$F_{(3,26)} = 7.7$, $p = 0.01$] and a significant interaction between treatment and time [$F_{(3,26)} = 7.81$, $p = 0.001$], but no effect of time [$F_{(3,26)} = 3.35$, $p = 0.08$] in the relative gene expression of *syt* in the Dl, with control fish at three weeks showing a higher expression, compared to all other groups (figure 2*a*). There were no effects of time [$F_{(2,31)} = 0.92$, $p = 0.34$] nor treatment [$F_{(2,31)} = 0.11$, $p = 0.74$] in the Dm (figure 2*b*). However, there was a significant effect of treatment [$F_{(2,28)} = 6.06$, $p = 0.02$], with exercised fish showing an overall higher *syt* expression than controls, but not of time [$F_{(2,28)} = 0.26$, $p = 0.61$] in the Vv (figure 2*c*).

### 3.2.2. Brain-derived neurotrophic factor

There were no effects of time [$F_{(2,32)} = 0.11$, $p = 0.74$; $F_{(2,32)} = 1.76$, $p = 0.19$] nor treatment [$F_{(2,32)} = 0.63$, $p = 0.43$; $F_{(2,32)} = 0.05$, $p = 0.83$] in the Dl (figure 2*d*) and Dm (figure 2*e*), respectively. There was a significant effect of treatment [$F_{(2,32)} = 4.22$, $p = 0.05$], with exercised fish showing an overall *bdnf* higher expression than controls, but not of time [$F_{(2,32)} = 0.42$, $p = 0.52$] in the Vv (figure 2*f*).

### 3.2.3. Proliferating nuclear cell antigen

There was a significant effect of treatment [$F_{(2,25)} = 5.15$, $p = 0.03$], with exercised fish showing an overall higher expression than controls, but not of time [$F_{(2,25)} = 0.02$, $p = 0.9$; figure 3*a*] in the Dl. There were no effects of time [$F_{(2,23)} = 0.61$, $p = 0.44$; $F_{(2,26)} = 0.28$, $p = 0.6$] nor treatment [$F_{(2,23)} = 0.2$, $p = 0.66$; $F_{(2,23)} = 0.54$, $p = 0.47$] in the Dm (figure 3*b*) and Vv (figure 3*c*), respectively.

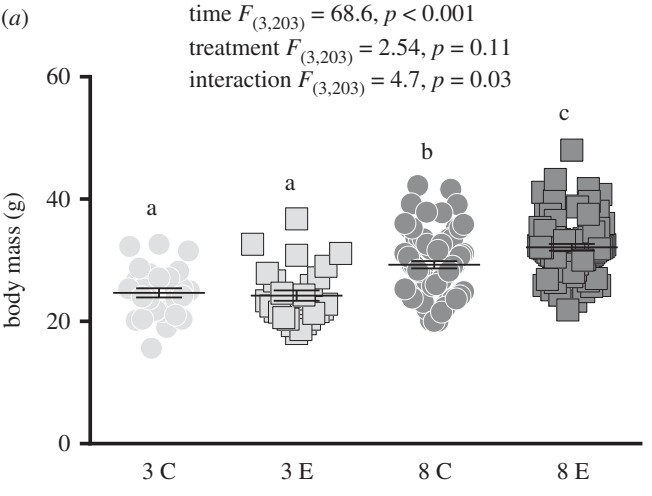

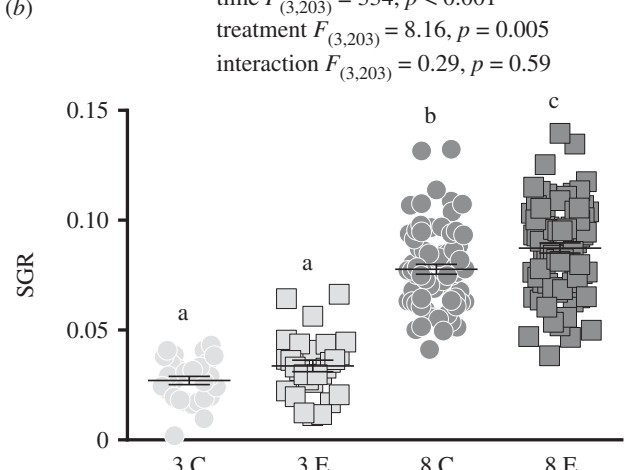

**Figure 1.** Body mass and specific growth rate (SGR) for exercised (E) and control (C) fish at three and eight weeks. Repeated-measures ANOVA statistics are given within each panel. Small letters symbolize statistical differences obtained with Tukey post hoc test. Values are given as mean ± s.e.m.

### 3.2.4. Neural differentiation factor d

There was a significant effect of time [$F_{(2,31)} = 4.75$, $p = 0.04$], with fish at three weeks showing an overall higher expression than groups at eight weeks, but not of treatment [$F_{(2,31)} = 1.98$, $p = 0.17$; figure 3e] in the Dm. There were no effects of time [$F_{(2,31)} = 0.15$, $p = 0.7$; $F_{(2,28)} = 2.15$, $p = 0.15$] nor treatment [$F_{(2,31)} = 2.77$, $p = 0.12$; $F_{(2,32)} = 2.11$, $p = 0.16$] in the Dl (figure 3d) and Vv (figure 3f), respectively.

## 3.3. RNA-seq

In order to determine whether exercise affects brain plasticity at the cellular level, we measured gene expression levels in the telencephalon of five fish per group using Illumina RNA-seq. We obtained between 10.1 and 39.1 million reads per sample (median 17 million), of which 90.6–95.8% (median 95.4%) was aligned to the salmon genome, of which, 63.2–79.1% (median 76.9%) could be attributed to a protein-coding gene.

An MDS plot of all samples (figure 4) shows a clear clustering of each experimental group, indicating robust gene expression changes correlated with the treatments. We therefore analysed differential expression of 27 171 genes (which does not include genes below a very low expression threshold of 10 reads per sample; figure 5). The contrast between exercise and control groups yielded 1772 genes differentially expressed using a FDR cut-off of 1%, of which 923 had significantly higher expression in

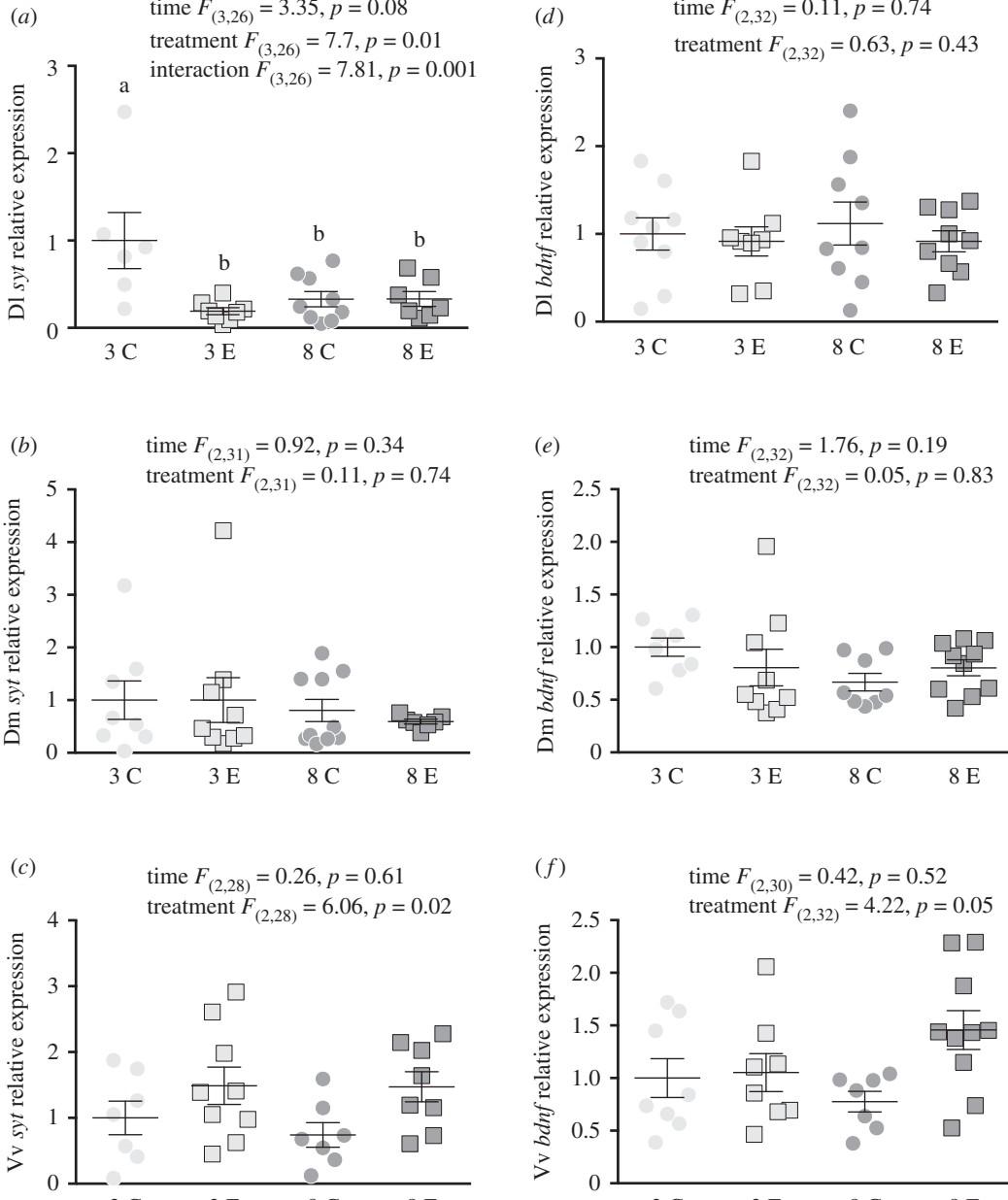

**Figure 2.** Mean (±s.e.m.) relative mRNA expression (to the *s20* reference gene) of the neuroplasticity markers *synaptotagmin* (*syt*) and *brain-derived neurotrophic factor* (*bdnf*) in the dorsolateral pallium (Dl), dorsomedial pallium (Dm) and the ventral part of the ventral telencephalon (Vv) of exercised (E) and control (C) fish at three and eight weeks. Statistics are given for each panel in the figure. Small letters symbolize statistical differences obtained with Tukey post hoc test.

swimmers (electronic supplementary material, table S3), and 849 had significantly higher expression in control fish (electronic supplementary material, table S4). A selection of these genes is presented in table 1.

## 3.4. Functional overrepresentation

To summarize which biological processes are over- or under-expressed in the swimmer group, we performed a GO category overrepresentation test on the sets of significantly differentially expressed genes between exercised fish and their controls. In total, 194 (electronic supplementary material, table S5) and 271 (electronic supplementary material, table S6) GO categories showed a significant overrepresentation of upregulated and downregulated genes, respectively ($p < 0.05$). The GO categories which related to neuroplasticity, neurogenesis or behavioural pathways involved with cognition were selected and relevant GO categories with a significant overrepresentation of

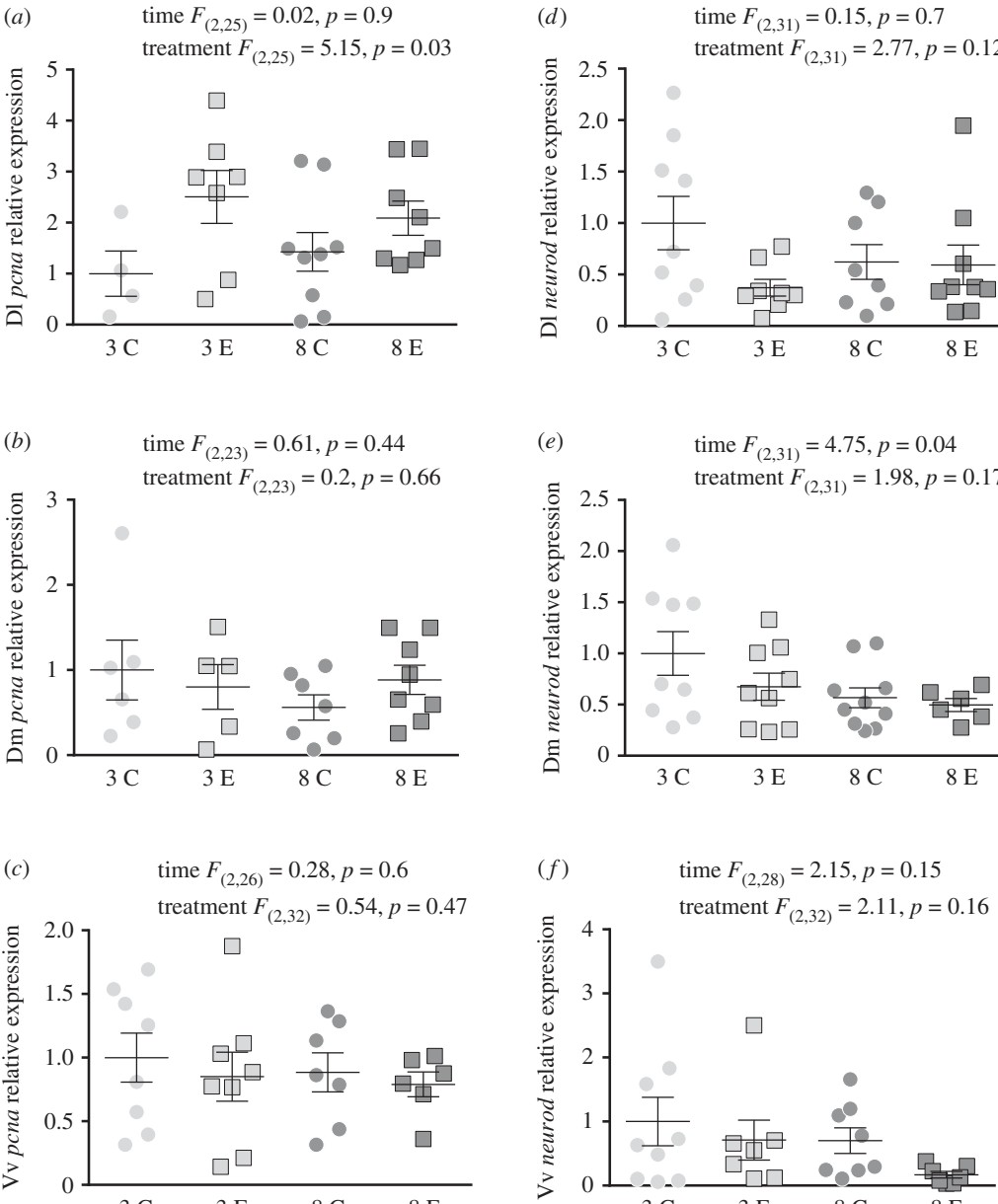

**Figure 3.** Mean (±s.e.m.) relative mRNA expression (to the *s20* reference gene) of the neurogenesis marker *proliferating cell nuclear antigen* (*pcna*) and the cell differentiation marker *neurotrophic factor d* (*neurod*) in the dorsolateral pallium (Dl), dorsomedial pallium (Dm) and the ventral part of the ventral telencephalon (Vv) of exercised (E) and control (C) fish at three and eight weeks. Statistics are given for each panel in the figure.

upregulated genes are presented in table 2, while downregulated GO categories are presented in table 3. The heat map in figure 6 provides a visual overview of the expression of selected neuroplasticity-related genes in all samples, showing a clustering of expression by treatment.

## 4. Discussion

We here report that sustained swimming exercise increases the expression of neuroplasticity- and cell proliferation-related genes in the telencephalon transcriptome of juvenile Atlantic salmon. Specifically, we report an upregulation of synaptic plasticity and neurogenesis, as well as downregulation of apoptosis genes in the whole telencephalon in response to exercise. In addition, there were region-specific differences in the expression of the neurogenesis marker *pcna* at both three and eight weeks, with higher expression in exercised fish in the Dl, which is the functional equivalent of the mammalian hippocampus.

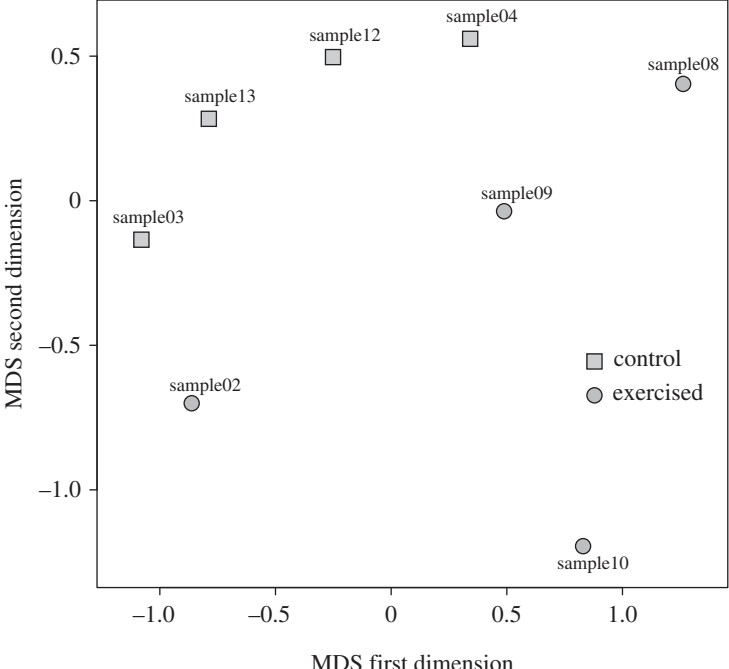

**Figure 4.** Multi-dimensional scaling (MDS) plot based on the expression of 27 171 genes. The first two dimensions clearly separate samples by treatment contrast (exercised versus control).

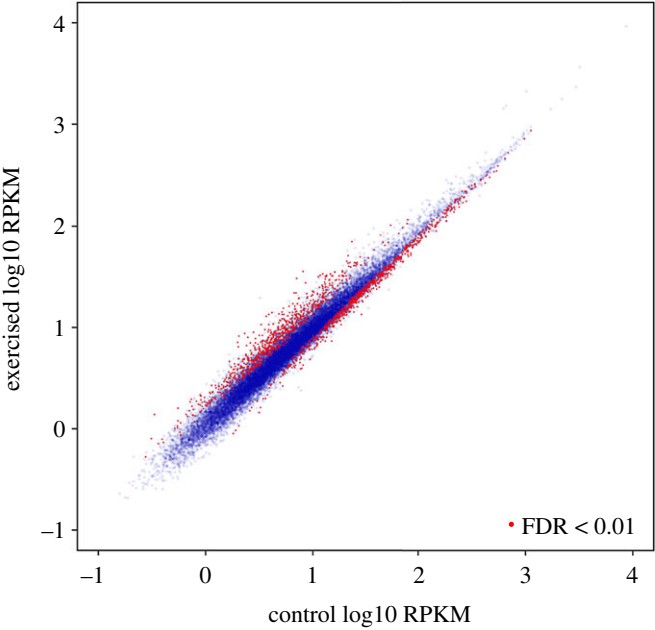

**Figure 5.** Forebrain gene expression in exercised versus control salmon. Depicted are expression values of 27 171 genes, normalized for between-sample differences in sequencing depth and within-sample transcript length differences (RPKM, reads per kilobase per million). Genes highlighted in red are significantly differentially expressed between exercised and control groups.

Furthermore, the expression of the neural plasticity markers *syt* and *bdnf* was also increased following exercise in the Vv, which is the functional equivalent to the mammalian lateral septum. Interestingly, exercised-induced increased neural plasticity in the lateral septum area has not been reported in mammalian studies and allows for novel target systems in the study of mechanisms associated with swimming, neural remodelling and cognition in fish as a model for other vertebrate systems.

The neurotrophin BDNF is a well-characterized neural growth factor which is important for synaptic plasticity and neural survival [4,31,32]. In exercised mammals, BDNF shows a robust upregulation in the

**Table 1.** Selection of significantly (false discovery rate (FDR) less than 0.01) differentially expressed genes in exercised fish, compared to unexercised controls. Expression is given as fold-change (FC) difference in exercised:control individuals.

| product | gene name | FC |
| --- | --- | --- |
| synaptic trafficking | | |
| synaptotagmin | | |
| synaptotagmin-7-like | LOC106592751 | 3.11 |
| synaptotagmin-11-like | LOC106569883 | 2.06 |
| synaptotagmin XVII | syt17 | 1.97 |
| synaptotagmin-7-like | LOC106587514 | 1.67 |
| synaptotagmin-7-like | LOC106562326 | 1.42 |
| syntaxin | | |
| syntaxin binding protein 5 (tomosyn) | stxbp5 | 1.71 |
| syntaxin-12-like | LOC106578562 | 0.78 |
| Signal transduction | | |
| CAM kinases | | |
| calcium/calmodulin-dependent protein kinase type II subunit beta-like | LOC106570801 | 2.23 |
| calmodulin-binding transcription activator 2-like | LOC106565081 | 1.97 |
| calcium/calmodulin-dependent protein kinase kinase 1-like | LOC106603934 | 1.94 |
| calcium/calmodulin-dependent protein kinase type 1-like | LOC106583531 | 1.91 |
| striatin%2C calmodulin-binding protein 4 | strn4 | 1.66 |
| calcium/calmodulin-dependent protein kinase type II subunit beta-like | LOC106569240 | 1.50 |
| calcium/calmodulin-dependent protein kinase kinase 2%2C beta | camkk2 | 1.37 |
| MAP kinases | | |
| mitogen-activated protein kinase 6-like | LOC106587736 | 1.67 |
| mitogen-activated protein kinase kinase kinase 12 | map3k12 | 1.63 |
| mitogen-activated protein kinase kinase kinase kinase 3-like | LOC106577155 | 1.61 |
| mitogen-activated protein kinase-activated protein kinase 5 | mapkapk5 | 1.34 |
| mitogen-activated protein kinase 6-like | LOC106594832 | 1.33 |
| protein kinase C | | |
| protein kinase C-binding protein NELL1-like | LOC106561902 | 1.66 |
| protein kinase C beta type | LOC106594520 | 1.66 |
| CREB | | |
| CREB-regulated transcription coactivator 3-like | LOC106562618 | 1.85 |
| CREB-binding protein-like | LOC106589530 | 1.42 |
| glutamatergic system | | |
| metabotropic glutamate receptor 5-like | LOC106563916 | 3.89 |
| metabotropic glutamate receptor 5-like | LOC106581415 | 2.53 |
| glutamate receptor 2-like | LOC106595266 | 2.38 |
| glutamate receptor ionotropic, NMDA 2B-like | LOC106601156 | 2.05 |
| glutamate receptor ionotropic, AMPA 4 | gria4 | 1.52 |
| glutamate decarboxylase 2 | gad2 | 1.51 |
| glutamate receptor ionotropic, kainate 5-like | LOC106576347 | 1.45 |
| glutamate receptor ionotropic, delta-2-like | LOC106580171 | 1.28 |

**Table 1.** (*Continued.*)

| product | gene name | FC |
| --- | --- | --- |
| GABAergic system | | |
| gamma-aminobutyric acid receptor-associated protein-like 2 | LOC106587940 | 0.68 |
| gamma-aminobutyric acid receptor-associated protein | gbrap | 0.72 |
| gamma-aminobutyric acid receptor-associated protein-like 1 | grl1 | 0.73 |
| gamma-aminobutyric acid receptor-associated protein | LOC106602900 | 0.74 |
| gamma-aminobutyric acid receptor-associated protein-like 1 | LOC106576832 | 0.75 |
| gamma-aminobutyric acid receptor-associated protein-like | LOC106577557 | 0.77 |
| gamma-aminobutyric acid receptor subunit alpha-5-like | LOC106563719 | 0.77 |
| gamma-aminobutyric acid receptor subunit beta-2-like | LOC106603846 | 2.26 |
| gamma-aminobutyric acid type B receptor subunit 1-like | LOC106570681 | 2.24 |
| gamma-aminobutyric acid receptor subunit gamma-1-like | LOC106610931 | 2.04 |
| gamma-aminobutyric acid type B receptor subunit 2-like | LOC106570447 | 1.89 |
| gamma-aminobutyric acid receptor subunit beta-1-like | LOC106610929 | 1.42 |
| growth factors | | |
| IGF | | |
| insulin-like growth factor 1 receptor | igf1r | 2.38 |
| insulin-like growth factor 1 receptor | LOC106592162 | 1.88 |
| FGF | | |
| fibroblast growth factor receptor 2 | fgfr2 | 1.36 |
| fibroblast growth factor receptor substrate 2-like | LOC106561545 | 0.68 |

dentate gyrus of the hippocampus [33,34], as well as in other brain regions, such as the amygdala [35]. Furthermore, BDNF plays a key role in activating the signal transduction pathways which drive increased neural plasticity [33,36]. Interestingly, our results on region-specific gene expression in the Dl, Dm and Vv show that exercised fish show an upregulation of *bdnf* only in the Vv. BDNF synaptic signalling is mainly mediated by the TrkB receptor tyrosine kinase. The binding of BDNF to the TrkB receptor triggers the autophosphorylation of tyrosine and leads to the activation of signalling pathways [37]. Therefore, in order to elucidate further how the exercise-mediated increase in Bdnf may be affecting signalling pathways in fish, future studies should include the expression of both Bdnf and TrkB. Interestingly, we also found in the Vv an exercise-induced increase in synaptotagmin (*syt*). Synaptotagmin is a synaptic vesicle protein that is selectively required as a $Ca^{2+}$ sensor in the regulation of neurotransmitter release [38]. These results suggest that the Vv in fish is a target for exercise-induced changes in neuroplasticity and this may give exercised fish an advantage in stress coping and goal-oriented behaviour, an exciting possibility that should be further investigated. Curiously, we found that control fish at three weeks had the highest expression of *syt* in the Dl. The reasons for this difference are not clear to us, particularly since this group was exposed to standard rearing conditions and this effect is not present at eight weeks or in any other brain area or marker.

Exercised fish showed significantly enhanced growth rates after eight weeks of training, compared to unexercised controls. Thus, our observation of enhanced growth in exercised fish suggests that our experimental treatment had beneficial effects on the physiology of the fish and did not lead to chronic stress. It can be argued that due to our experimental set-up, it is not possible to control for any potential tank effects affecting the results obtained in growth and gene expression. However, the result of enhanced growth due to increased swimming exercise in this study is in agreement with previous studies on salmonids (e.g. [17]). Furthermore, preliminary data from a pilot experiment we have conducted in a different experimental set-up shows that the telencephalic gene expression profile of fish subjected to a forced-exercise training regime is similar to the one reported in this study.

After eight weeks of swimming, 1772 transcripts in the telencephalon were differentially expressed in exercised fish compared to unexercised controls. GO analysis attributed these differences in transcript abundance to increased expression of genes associated with processes relating to neural plasticity,

**Table 2.** Selection of gene ontology (GO) categories with a significant ($p < 0.05$) overrepresentation of upregulated genes in exercised fish, sorted by theme.

| theme | GO ID | gene ontology term | DE in category | total genes in category | *p*-value |
|---|---|---|---|---|---|
| np | GO:1990090 | cellular response to nerve growth factor stimulus | 10 | 131 | 0.0375 |
| np | GO:0050775 | positive regulation of dendrite morphogenesis | 8 | 80 | 0.0399 |
| np | GO:0048172 | regulation of short-term neuronal synaptic plasticity | 9 | 79 | 0.0212 |
| np | GO:0050803 | regulation of synapse structure or activity | 11 | 72 | 0.0010 |
| np | GO:1903861 | positive regulation of dendrite extension | 9 | 69 | 0.0330 |
| np | GO:0051963 | regulation of synapse assembly | 7 | 54 | 0.0452 |
| np | GO:0060996 | dendritic spine development | 5 | 48 | 0.0239 |
| np | GO:0070983 | dendrite guidance | 9 | 30 | 0.0034 |
| np | GO:0051387 | negative regulation of neurotrophin TRK receptor signalling pathway | 4 | 26 | 0.0410 |
| cp | GO:2000648 | positive regulation of stem cell proliferation | 6 | 40 | 0.0089 |
| cp | GO:0010458 | exit from mitosis | 6 | 39 | 0.0122 |
| cp | GO:0045927 | positive regulation of growth | 5 | 25 | 0.0037 |
| cp | GO:0070317 | negative regulation of G0 to G1 transition | 4 | 20 | 0.0209 |
| cp | GO:0097193 | intrinsic apoptotic signalling pathway | 8 | 128 | 0.0225 |
| cp | GO:0042771 | intrinsic apoptotic signalling pathway in response to DNA damage by p53 class mediator | 7 | 74 | 0.0467 |
| cp | GO:0072577 | endothelial cell apoptotic process | 6 | 20 | 0.0008 |
| cp | GO:0042981 | regulation of apoptotic process | 21 | 245 | 0.0013 |
| cp | GO:0008285 | negative regulation of cell proliferation | 47 | 855 | 0.0325 |
| cp | GO:0007052 | mitotic spindle organization | 13 | 181 | 0.0146 |
| cp | GO:0000088 | mitotic prophase | 9 | 138 | 0.0327 |
| cp | GO:0090307 | mitotic spindle assembly | 7 | 113 | 0.0479 |
| cp | GO:0051225 | spindle assembly | 9 | 78 | 0.0086 |
| beh | GO:0007611 | learning or memory | 13 | 138 | 0.0254 |
| beh | GO:0050890 | cognition | 8 | 66 | 0.0374 |

np, neuroplasticity; cp, cell proliferation; beh, behaviour; DE, differentially expressed.

such as dendritic spine development, synaptic plasticity and cell proliferation, as well as downregulation of genes associated with apoptosis in exercised individuals, compared to control fish. This is in agreement with mammalian studies, which report that exercise stimulates the forebrain, particularly the hippocampus, through the increased abundance of neurotrophins such as BDNF, as well as stimulation of memory and learning-related processes such as long-term potentiation [39,40]. Recent

**Table 3.** Selection of gene ontology (GO) categories with a significant ($p < 0.05$) overrepresentation of downregulated genes in exercised fish.

| theme | GO ID | gene ontology term | DE in category | total genes in category | p-value |
|---|---|---|---|---|---|
| np | GO:0048812 | neuron projection morphogenesis | 9 | 162 | 0.021 |
| np | GO:0061001 | regulation of dendritic spine morphogenesis | 6 | 62 | 0.021 |
| np | GO:0008582 | regulation of synaptic growth at neuromuscular junction | 4 | 39 | 0.000 |
| np | GO:0030182 | neuron differentiation | 11 | 152 | 0.034 |
| np | GO:0010976 | positive regulation of neuron projection development | 16 | 356 | 0.007 |
| cp | GO:0006915 | apoptotic process | 64 | 913 | 0.000 |
| cp | GO:0010940 | positive regulation of necrotic cell death | 5 | 28 | 0.001 |
| cp | GO:0097193 | intrinsic apoptotic signalling pathway | 11 | 128 | 0.006 |
| cp | GO:0070265 | necrotic cell death | 5 | 23 | 0.000 |
| cp | GO:0008625 | extrinsic apoptotic signalling pathway via death domain receptors | 5 | 76 | 0.042 |
| cp | GO:0042981 | regulation of apoptotic process | 14 | 245 | 0.002 |
| cp | GO:1900740 | positive regulation of protein insertion into mitochondrial membrane involved in apoptotic signalling pathway | 10 | 74 | 0.000 |
| cp | GO:2001243 | negative regulation of intrinsic apoptotic signalling pathway | 6 | 60 | 0.018 |
| cp | GO:0022008 | neurogenesis | 25 | 291 | 0.002 |
| cp | GO:0030307 | positive regulation of cell growth | 13 | 225 | 0.035 |
| cp | GO:0048680 | positive regulation of axon regeneration | 5 | 23 | 0.002 |
| cp | GO:0051437 | positive regulation of ubiquitin-protein ligase activity involved in the regulation of mitotic cell cycle transition | 24 | 162 | 0.000 |
| cp | GO:0000090 | mitotic anaphase | 27 | 327 | 0.011 |
| cp | GO:0007052 | mitotic spindle organization | 18 | 181 | 0.000 |
| cp | GO:0007346 | regulation of mitotic cell cycle | 7 | 132 | 0.042 |
| cp | GO:0090307 | mitotic spindle assembly | 10 | 113 | 0.018 |
| cp | GO:0007088 | regulation of mitotic nuclear division | 7 | 89 | 0.009 |
| cp | GO:0008156 | negative regulation of DNA replication | 4 | 37 | 0.047 |

np, neuroplasticity; cp, cell proliferation; DE, differentially expressed.

studies have proposed the molecular pathways underlying exercise-induced neurogenesis and synaptic plasticity (reviewed by Lista & Sorrentino [36]). In summary, physical activity in mammals first leads to an increased abundance of neurotrophins such as BDNF and insulin-like growth factor (IGF). Subsequently, BDNF can directly promote neurogenesis, or it may activate signal transduction pathways through signalling molecules, such as calcium/calmodulin-dependent protein kinase II (CAMK-II), mitogen-activated protein kinase (MAPK), protein kinase C (PKC) and cAMP response element binding (CREB) protein, which in turn stimulate neural processes such as synaptogenesis and LTP. Furthermore, synaptogenesis is enhanced by synaptic trafficking molecules such as synaptotagmin and syntaxin, which are promoted through CAMK-II after activation by BDNF or IGF. Notably, our forebrain transcriptome gene expression profile analysis in exercised fish uncovered an upregulation of several genes within these pathways, such as synaptotagmin, syntaxin CAMK-II,

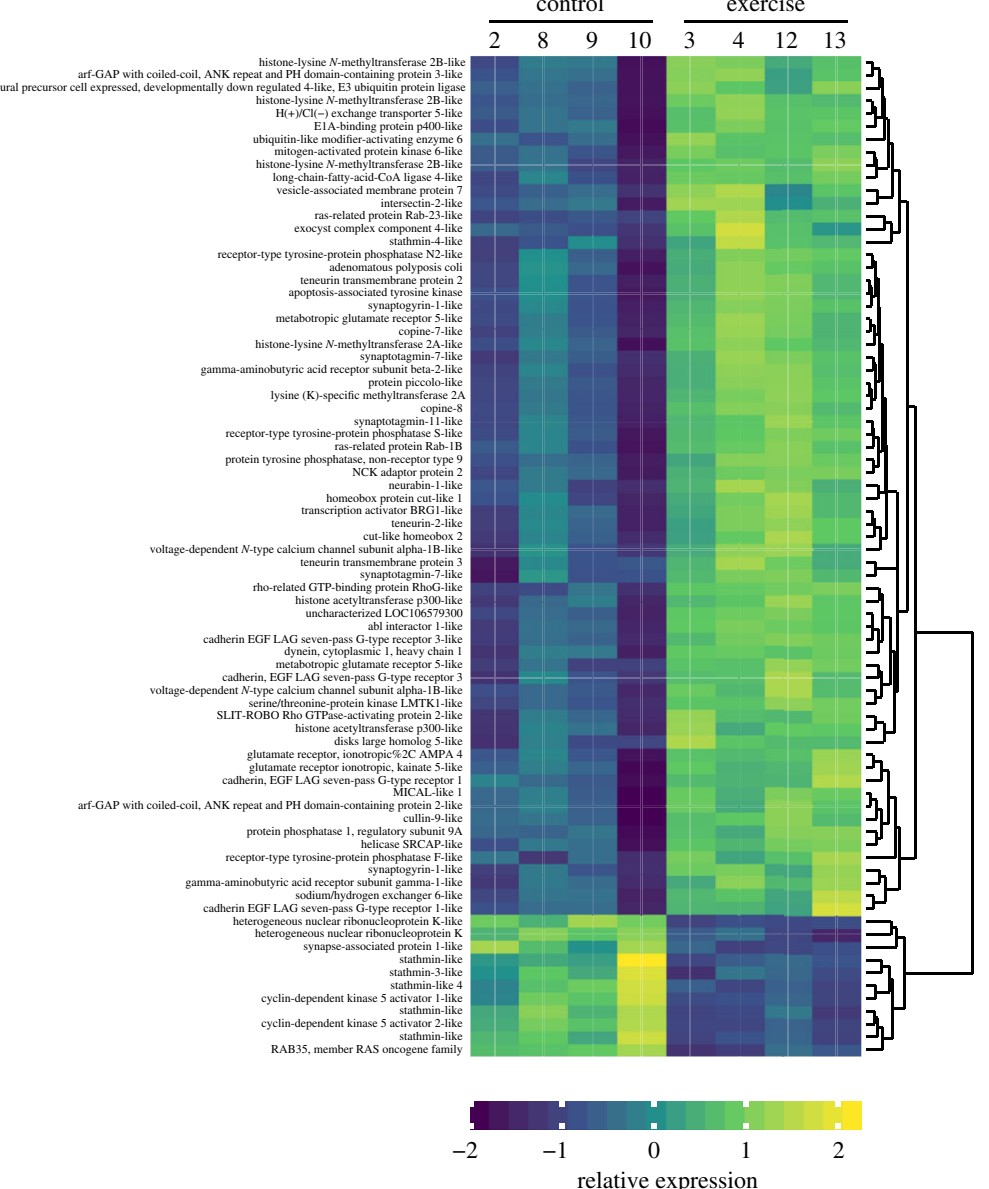

**Figure 6.** Heat map depicting expression of differentially expressed genes within gene ontology (GO) categories related to neuroplasticity. In order to make the absolute expression levels and amplitudes of expression changes comparable between genes, for every gene the original read per kilobase million (RPKM) values were converted to *z*-scores (i.e. expressed in standard deviations around the mean). Hierarchical clustering based on Pearson correlation was used to arrange genes by similarity in expression pattern.

MAPK, PKC and CREB, as well as two IGF receptor-related transcripts. Molteni *et al.* [33] reported that running exercise activates the mammalian hippocampal glutamatergic system and suppresses the gamma-aminobutyric acid (GABA)ergic system. Similarly, exercised fish in our experiment showed increased expression of several glutamate receptor transcripts and reduced expression of several GABA receptor transcripts, although effects on the GABAergic system are somewhat ambivalent, as we observed concurrent upregulation of several GABA receptor subunit transcripts in exercised individuals. In summary, there appear to be several parallels between the teleostean and mammalian neural response to exercise in processes regarding synaptic trafficking, signal transduction and the glutamatergic and GABAergic systems. These findings suggest that many of the molecular pathways which have been proposed to underlie exercise-induced neuroplasticity are conserved between mammals and teleost fish.

The transcriptome of exercised fish revealed a significant overrepresentation of downregulated genes in several GO categories related to apoptosis. This is an interesting observation, as mammalian studies

have revealed that, besides promoting neurogenesis, exercise also increases cell survival [41] and can inhibit neuronal apoptosis, particularly in ageing animals [42] or individuals with traumatic brain injury [43]. Our observation that swimming exercise may also affect neuronal apoptosis in fish suggests that fish models may be particularly amenable in the study of exercise-induced neural repair in damaged animals, as fish with neural damage show an incredible capacity for neural regeneration in the central nervous system [6]. A better understanding of the effects of swimming exercise on neuroplasticity in fish species may have further ramifications for human disease research, as an important application of mammalian exercise-induced neuroplasticity is its potential to prevent cognitive decline, particularly in the context of ageing and neurodegenerative diseases (reviewed by Ma *et al.* [3]).

Increased neurogenesis in response to exercise in mammals has been reported in the hippocampus and is associated with increased capacity for spatial tasks [2–4]. In agreement with these results, we found an increased *pcna* expression (a commonly used marker for neurogenesis, e.g. [44]) in the Dl, which is functionally equivalent to the mammalian hippocampus [13,45] and plays an important role in spatial memory in fish [46,47]. It is of particular interest that Luchiari & Chacon [7] report an improvement in learning in goal-oriented behaviour (i.e. classical conditioning) in zebrafish subjected to increased swimming exercise. Therefore, future studies should combine behaviour and brain region-specific studies in order to elucidate exercise-induced effects in fish. Interestingly, an increase in the expression of the neurogenesis marker *pcna* was not accompanied by increased expression of the cell differentiation marker *neurod*. In fact, we found that *neurod* expression decreased over time in the Dm of both groups, with the lowest values found at eight weeks. Neurod is a transcription factor involved in the differentiation of cells into maturation [48]. The lack of expression of this gene at the sampled timepoints suggests that exercise does not lead to increased cell differentiation, at least within the timeframe and within the forebrain areas that we studied. In other words, the exercise-induced neurogenesis indicated by increased *pcna* expression may represent a recent increase in new-born cells which have not yet gone through differentiation and maturation. Importantly, while expression of neuroplasticity-associated genes are commonly used as markers for neuroplasticity (for reviews, see Zupanc & Lamprecht [49]), it is important to note that mRNA gene expression does not always mirror protein levels. We therefore emphasize that future research on exercised-induced neurogenesis in fish species should corroborate the findings of this study using additional techniques for visualizing neurogenesis, such as 5-ethynyl-2′-deoxyuridine (EdU) labelling and Pcna, NeuroD, Bdnf and TrkB immunohistochemistry.

In conclusion, we report that juvenile fish show an exercise-induced increased expression of neuroplasticity and neurogenesis markers in the forebrain. We are among the first to study the effects of exercise on neuroplasticity in fish and our results uncover several parallels with mammalian studies, such as increased *pcna* expression in functional equivalent of the hippocampus in fish (i.e. the Dl) and increased expression of neuroplasticity, synaptic trafficking and signal transduction in the telencephalon, with special emphasis in the fish's lateral septum functional equivalent (i.e. the Vv). Future studies should study the link between exercised-induced neural plasticity and cognition in fish. In addition, histological studies looking at protein abundance of neurogenesis and neural plasticity markers should further elucidate the region-specific effects of exercise training in all regions of the fish brain. These endeavours will help shed light on possible applications of increased swimming training in husbandry practices aimed at increasing animal welfare (aquaculture), as well as validating fish as an ideal model in the study of vertebrate neural mechanisms, including the functional relationship between exercise and neurodegenerative diseases.

Ethics. This experiment was performed in accordance with Dutch law for experimentation and procedures on live animals. The experimental protocol was approved by the Animal Experimental Committee (DEC) of Wageningen University and Research (case no. 2016.D-0039).

Data availability. All relevant data are within the paper or published as electronic supplementary material.

Authors' contributions. D.M. designed and conducted the study, conducted sampling and processed samples used in RNAseq, participated in biometrical data analysis and helped with RNAseq data analysis, helped draft the manuscript. A.P.P. designed and coordinated the study, helped with biometric data analysis, helped draft the manuscript. I.M. coordinated study, helped with sample collection and helped draft the manuscript. C.V.H. conducted all RNAseq data analysis and helped draft the manuscript. M.A.V. designed the study, processed microdissection samples, conducted all molecular biology analyses, analysed final datasets and drafted the manuscript. All authors gave final approval for publication.

Competing interests. The authors declare no competing or financial interests.

Funding. This research was funded through the European Union's Horizon 2020 research and innovation programme under grant agreement no. 652831 (AQUAEXCEL[2020]) and under the Marie Sklodowska-Curie Actions: Innovative Training Network 'IMPRESS', grant agreement no. 642893. Personal travel grants to D.M. were provided by COST Action FA1304 'Swimming of fish and implications for migration and aquaculture' (FITFISH) and the Research Council of Norway under the HAVBRUK programme, project no. 168075/E40.

Acknowledgements. The authors thank Menno ter Veld, Wian Nusselder, Emily Roux, Sander Visser, Truus van der Wal and Amerik Schuitemaker (CARUS, Wageningen University and Research) for assistance with animal care.

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
