## [Reviewer comments · Royal Society Open Science]

Review History

RSOS-191640.R0 (Original submission)

Review form: Reviewer 1

Is the manuscript scientifically sound in its present form?

Yes

Are the interpretations and conclusions justified by the results?

Yes

Is the language acceptable?

Yes

Do you have any ethical concerns with this paper?

No

Have you any concerns about statistical analyses in this paper?

No

Recommendation?

Accept with minor revision (please list in comments)

Comments to the Author(s)

This is a very interesting study addressing the effects of exercise on neuronal plasticity. The results show that exercise increases the expression of genes related to neurogenesis and plasticity and at the same time reduces expression of genes related to apoptotic pathways. These are clearly novel findings, at least in teleost fish. This is also pointed out by the authors already in the abstract and throughout the manuscript. It is perhaps enough to state this novelty once. The first sentence in the abstract states that it is well established in mammals that running exercise enhances brain plasticity and cognitive performance but that this phenomenon has not been much studied in fish. First, I don't think that this effect in mammals is restricted to running but is a more general response to endurance exercise. Moreover, fish do not run. Thus I would suggest that running exercise is exchanged for endurance exercise. As I understand, only one exercise and one control tank was used in the experiment. This is a potential problem since tank effects are common and could not be controlled for. This needs to be acknowledged in the discussion.

Review form: Reviewer 2

Is the manuscript scientifically sound in its present form?

Yes

Are the interpretations and conclusions justified by the results?

Yes

Is the language acceptable?

Yes

Do you have any ethical concerns with this paper?

No

Have you any concerns about statistical analyses in this paper?

No

Recommendation?

Accept with minor revision (please list in comments)

Comments to the Author(s)

The paper by Mes and colleagues characterizes a peculiar aspect on the potential beneficial effects of swimming exercise in juvenile specimens of Atlantic Salmon. The approach and methodology sound correct, some minor concerns need to be addressed:

1. were authors able to distinguish between male and female?
2. did they observe preference place in the tank (bottom or wall or water surface) in course of swimming exercise?
3. how many animals were used for RNA sequencing?
4. it is really interesting to see that bdnf appears upregulated in the Vv and not in the homologous of mammalian hippocampus, where it is quite abundant in other teleost species (mainly zebrafish). To better unravel the reason of this upregulation, have the authors checked also for the specific receptor TrkB? Based on these results, it could be expected to see also an upregulation of TrkB, reinforcing the hypothesis that in course of swimming exercise the homologous of lateral septum and not hippocampus is much more involved.

5. authors do not discuss the unchanged levels of NeuroD over the entire experiment and telencephalic region. Can they comment on this?

RESULTS:

paragraph gene expression:

- Fig. 2B refers to Fig 2C and viceversa.
- Better to mention figures in order (2D,2E,2F) as they appear in the figure

DISCUSSION:

Would be better to make a unique discussion on bdnf upregulation in the homologous of mammalian lateral septum. Therefore, move the paragraph from line 319 to 332 after the paragraph at the line 273. As it sounds quite repetitive.

Line 241 change aligning >>aligning

Decision letter (RSOS-191640.R0)

22-Nov-2019

Dear Dr Vindas,

On behalf of the Editors, I am pleased to inform you that your Manuscript RSOS-191640 entitled "Swimming exercise enhances brain plasticity in fish" has been accepted for publication in Royal Society Open Science subject to minor revision in accordance with the referee suggestions. Please find the referees' comments at the end of this email.

The reviewers and handling editors have recommended publication, but also suggest some minor revisions to your manuscript. Therefore, I invite you to respond to the comments and revise your manuscript.

- Ethics statement

- Data accessibility

<http://datadryad.org/submit?journalID=RSOS&manu=RSOS-191640>

- **Competing interests**

- **Authors' contributions**

- **Acknowledgements**

- **Funding statement**

Because the schedule for publication is very tight, it is a condition of publication that you submit the revised version of your manuscript before 01-Dec-2019. Please note that the revision deadline will expire at 00.00am on this date. If you do not think you will be able to meet this date please let me know immediately.

If your manuscript is newly submitted and subsequently accepted for publication, you will be asked to pay the article processing charge, unless you request a waiver and this is approved by Royal Society Publishing. You can find out more about the charges at <https://royalsocietypublishing.org/rsos/charges>. Should you have any queries, please contact openscience@royalsociety.org.

Kind regards,

on behalf of Dr Susana Lopes (Associate Editor) and Kevin Padian (Subject Editor)
openscience@royalsociety.org

Associate Editor Comments to Author (Dr Susana Lopes):

Based on the reviewers comments I recommend the manuscript is accepted with minor changes. These changes need to be included in the revised text and explained in the cover letter with line tracking.

Special attention must be given to the Discussion section. This needs to be improved and include the requirements from the reviewers. Please read their comments carefully and:

- 1- Discuss the expression of TrkB, relating it to the hypothesis that in course of swimming exercise the homologous of lateral septum and not hippocampus is much more involved.
- 2- Discuss the unchanged levels of NeuroD.

Reviewer comments to Author:

Reviewer: 1

Comments to the Author(s)

This is a very interesting study addressing the effects of exercise on neuronal plasticity. The results show that exercise increases the expression of genes related to neurogenesis and plasticity and at the same time reduces expression of genes related to apoptotic pathways. These are clearly novel findings, at least in teleost fish. This is also pointed out by the authors already in the abstract and throughout the manuscript. It is perhaps enough to state this novelty once. The first sentence in the abstract states that it is well established in mammals that running exercise enhances brain plasticity and cognitive performance but that this phenomenon has not been much studied in fish. First, I don't think that this effect in mammals is restricted to running but is a more general response to endurance exercise. Moreover, fish do not run. Thus I would suggest that running exercise is exchanged for endurance exercise. As I understand, only one exercise and one control tank was used in the experiment. This is a potential problem since tank effects are common and could not be controlled for. This needs to be acknowledged in the discussion.

Reviewer: 2

Comments to the Author(s)

The paper by Mes and colleagues characterizes a peculiar aspect on the potential beneficial effects of swimming exercise in juvenile specimens of Atlantic Salmon. The approach and methodology sound correct, some minor concerns need to be addressed:

1. were authors able to distinguish between male and female?
2. did they observe preference place in the tank (bottom or wall or water surface) in course of swimming exercise?
3. how many animals were used for RNA sequencing?
4. it is really interesting to see that bdnf appears upregulated in the Vv and not in the homologous of mammalian hippocampus, where it is quite abundant in other teleost species (mainly zebrafish). To better unravel the reason of this upregulation, have the authors checked

also for the specific receptor TrkB? Based on these results, it could be expected to see also an upregulation of TrkB, reinforcing the hypothesis that in course of swimming exercise the homologous of lateral septum and not hippocampus is much more involved.

5. authors do not discuss the unchanged levels of NeuroD over the entire experiment and telencephalic region. Can they comment on this?

RESULTS:

paragraph gene expression:

- Fig. 2B refers to Fig 2C and viceversa.

- Better to mention figures in order (2D,2E,2F) as they appear in the figure

DISCUSSION:

Would be better to make a unique discussion on bdnf upregulation in the homologous of mammalian lateral septum. Therefore, move the paragraph from line 319 to 332 after the paragraph at the line 273. As it is sounds quite repetitive.

Line 241 change aligning >>aligning

Author's Response to Decision Letter for (RSOS-191640.R0)

See Appendix A.

Decision letter (RSOS-191640.R1)

04-Dec-2019

Dear Dr Vindas,

It is a pleasure to accept your manuscript entitled "Swimming exercise enhances brain plasticity in fish" in its current form for publication in Royal Society Open Science. The comments of the reviewer(s) who reviewed your manuscript are included at the foot of this letter.

Please note that your colleague's email address daan.mes@nmbu.no is not currently accepting messages - please can you check this or supply an updated email address for us?

Due to rapid publication and an extremely tight schedule, if comments are not received, your paper may experience a delay in publication. Royal Society Open Science operates under a continuous publication model. Your article will be published straight into the next open issue and

this will be the final version of the paper. As such, it can be cited immediately by other researchers. As the issue version of your paper will be the only version to be published I would advise you to check your proofs thoroughly as changes cannot be made once the paper is published.

on behalf of Dr Susana Lopes (Associate Editor) and Kevin Padian (Subject Editor)
openscience@royalsociety.org

Appendix A

Dear Dr. Dr Susana Lopes,

Please find enclosed our revised manuscript “Swimming exercise enhances brain plasticity in fish” (RSOS-191640).

We are grateful for your, and the reviewers, evaluation of our work and would like to thank for instructive and helpful comments. We have given close attention to the issues raised and addressed these in our revised manuscript. Please find a detailed response to each comment below.

Sincerely,

Marco A. Vindas

Associate Editor Comments to Author (Dr Susana Lopes):

Based on the reviewers comments I recommend the manuscript is accepted with minor changes. These changes need to be included in the revised text and explained in the cover letter with line tracking.

Special attention must be given to the Discussion section. This needs to be improved and include the requirements from the reviewers. Please read their comments carefully and:

1. Discuss the expression of TrkB, relating it to the hypothesis that in course of swimming exercise the homologous of lateral septum and not hippocampus is much more involved.

We have now included the role of TrkB in BDNF signaling in Ln 283-287. We do not currently have gene expression levels for this receptor since developing the molecular tools necessary for this endeavor represents a project in itself and goes beyond the context of the current paper. Encouraged by the results obtained in this experiment we are in the process of developing several research tools that will be used in future projects for the further study into the mechanisms at play in swimming-enhanced neuroplasticity in fish

2. Discuss the unchanged levels of NeuroD.

A more extensive discussion of these results is now found in Ln 352-366

Reviewer 1. Comments to the Author(s)

1. This is a very interesting study addressing the effects of exercise on neuronal plasticity. The results show that exercise increases the expression of genes related to neurogenesis and plasticity and at the same time reduces expression of genes related to apoptotic pathways. These are clearly novel findings, at least in teleost fish. This is also pointed out by the authors already in the abstract and throughout the manuscript. It is perhaps enough to state this novelty once.

In the current version we mentioned the novelty of this experiment twice. *We feel that it is natural to state this in the abstract (Ln 23-26), but also in the conclusions (Ln 368-372) since we highlight these results in a broader context. We feel that this conveys a message in key areas of the manuscript.*

2. The first sentence in the abstract states that it is well established in mammals that running exercise enhances brain plasticity and cognitive performance but that this phenomenon has not been much studied in fish. First, I don't think that this effect in mammals is restricted to running but is a more general response to endurance exercise. Moreover, fish do not run. Thus, I would suggest that running exercise is exchanged for endurance exercise.

We thank the reviewer for pointing this out, we have now changed this sentence to read: "It is well-established that sustained exercise training can enhance brain plasticity and boost cognitive performance in mammals" (Ln 11-12). We feel that this is more accurate, and it is also consistent with statements made in the introduction.

3. As I understand, only one exercise and one control tank was used in the experiment. This is a potential problem since tank effects are common and could not be controlled for. This needs to be acknowledged in the discussion.

We include a discussion about this in the current manuscript Ln 298-303

Reviewer 2. Comments to the Author(s)

1. were authors able to distinguish between male and female?

We did distinguish between males and females for sampled fish only. However, there were no significant effects of sex on the studied parameters and we therefore did not include this information. However, we have now included a brief explanation about this in the statistical analyses section (Ln 196-199) and note that the fish gender is included in the supplementary data set file.

2. did they observe preference place in the tank (bottom or wall or water surface) in course of swimming exercise?

In general, we noticed that exercised fish showed no sign of fatigue and generally positioned themselves facing the current, while occasionally drifting down with the current (Ln 96-98). We also observed that individuals appear to preferentially position themselves in the same location throughout the experiment, while some preferred the stronger currents, others chose the weaker ones. These observations are only anecdotal and not quantified and we have therefore chosen not to include this information except for a brief mention of this in Ln 90-91: Thus, by positioning themselves throughout the tank, fish could 'choose' their preferred swimming speed.

3. how many animals were used for RNA sequencing?

We have included this information in Ln 160. We sampled 4 fish per treatment for RNAseq analysis.

4. it is really interesting to see that *bdnf* appears upregulated in the Vv and not in the homologous of mammalian hippocampus, where it is quite abundant in other teleost species (mainly zebrafish). To better unravel the reason of this upregulation, have the authors checked also for the specific receptor TrkB? Based on these results, it could be expected to see also an upregulation of TrkB, reinforcing the hypothesis that in course of swimming exercise the homologous of lateral septum and not hippocampus is much more involved.

The upregulation of bdnf in the Vv in this context is quite interesting. We agree with the reviewer that in order to understand the significance of this upregulation it would be beneficial to also include expression of the TrKb receptor. We have now included a short segment about this in our discussion (Ln 283-287). However, there is a lack of genetic tools regarding the specific sequence for TrKb in fish. Developing the tools in order to find specific primers for this gene were beyond the scope of the current work and we were not able to include this in our original exploratory analysis. Notably, there are a few immunohistochemical studies that have been able to quantify the protein expression of this receptor in fish (e.g. D'Angelo et al. 2012. Microscopy Res & Tech 75(1): 81-88). We believe therefore that the next logical step is to conduct further experiments that include immunohistochemistry analysis of both TrKb and Bdnf (as well as PcnA, NeuroD and BrdU) in order to elucidate further the swimming-induced neuroplasticity changes in fish neural region-specific areas (Ln 360-366)

5. authors do not discuss the unchanged levels of NeuroD over the entire experiment and telencephalic region. Can they comment on this?

We had briefly discussed these results in our previous manuscript version, but we have now elaborated our discussion of this topic in Ln 352-360

6. RESULTS: paragraph gene expression:

- Fig. 2B refers to Fig 2C and viceversa.
- Better to mention figures in order (2D,2E,2F) as they appear in the figure

Both suggestions have now been implemented in Ln 221-229

7. DISCUSSION: Would be better to make a unique discussion on *bdnf* upregulation in the homologous of mammalian lateral septum. Therefore, move the paragraph from line 319 to 332 after the paragraph at the line 273. As it is sounds quite repetitive.

The proposed change has been implemented. The paragraph is now in Ln 277-294

8. Line 241 change aligning >>aligning

We have now rewritten this sentence and the word aligning is not used anymore (Ln 244)